# Apheresis Efficacy and Tolerance in the Setting of HLA-Incompatible Kidney Transplantation

**DOI:** 10.3390/jcm10061316

**Published:** 2021-03-23

**Authors:** Johan Noble, Antoine Metzger, Hamza Naciri Bennani, Melanie Daligault, Dominique Masson, Florian Terrec, Farida Imerzoukene, Beatrice Bardy, Gaelle Fiard, Raphael Marlu, Eloi Chevallier, Benedicte Janbon, Paolo Malvezzi, Lionel Rostaing, Thomas Jouve

**Affiliations:** 1Nephrology, Hemodialysis, Apheresis and Kidney Transplantation Department, University Hospital Grenoble, 38000 Grenoble, France; jnoble@chu-grenoble.fr (J.N.); ametzger@ch-annecygenevois.fr (A.M.); hnaciribennani@chu-grenoble.fr (H.N.B.); mdailgault@chu-grenoble.fr (M.D.); fterrec@chu-grenoble.fr (F.T.); fimerzoukene@chu-grenoble.fr (F.I.); echevallier1@chu-grenoble.fr (E.C.); bjanbon@chu-grenoble.fr (B.J.); pmalvezzi@chu-grenoble.fr (P.M.); tjouve@chu-grenoble.fr (T.J.); 2University Grenoble Alpes, 38000 Grenoble, France; 3HLA Laboratory—Établissement Français du Sang-EFS-, 38000 Grenoble, France; dominique.masson@efs.sante.fr (D.M.); beatrice.bardy@efs.sante.fr (B.B.); 4Urology Department, University Hospital Grenoble, 38000 Grenoble, France; gfiard@chu-grenoble.fr; 5TIMC-IMAG, Grenoble INP, CNRS, University Grenoble Alpes, F-38000 Grenoble, France; 6Haemostasis Laboratory, University Hospital Grenoble, 38000 Grenoble, France; rmarlu@chu-grenoble.fr

**Keywords:** plasmapheresis, kidney transplantation, desensitization, donor specific antibody

## Abstract

Nearly 18% of patients on a waiting list for kidney transplantation (KT) are highly sensitized, which make access to KT more difficult. We assessed the efficacy and tolerance of different techniques (plasma exchanges [PE], double-filtration plasmapheresis [DFPP], and immunoadsorption [IA]) to remove donor specific antibodies (DSA) in the setting of HLA-incompatible (HLAi) KT. All patients that underwent apheresis for HLAi KT within a single center were included. Intra-session and inter-session Mean Fluorescence Intensity (MFI) decrease in DSA, clinical and biological tolerances were assessed. A total of 881 sessions were performed for 45 patients: 107 DFPP, 54 PE, 720 IA. The procedures led to HLAi KT in 39 patients (87%) after 29 (15–51) days. A higher volume of treated plasma was associated with a greater decrease of inter-session class I and II DSA (*p* = 0.04, *p* = 0.02). IA, PE, and a lower maximal DSA MFI were associated with a greater decrease in intra-session class II DSA (*p* < 0.01). Safety was good: severe adverse events occurred in 17 sessions (1.9%), more frequently with DFPP (6.5%) *p* < 0.01. Hypotension occurred in 154 sessions (17.5%), more frequently with DFPP (*p* < 0.01). Apheresis is well tolerated (IA and PE > DFPP) and effective at removing HLA antibodies and allows HLAi KT for sensitized patients.

## 1. Introduction

Chronic kidney disease (CKD) and end-stage kidney disease (ESKD) are global public health problems. Kidney transplantation (KT) provides the best results in terms of survival, quality of life, and health-care savings compared to hemodialysis (HD) when kidney replacement is necessary [1].

Currently, the major causes of restricting access to KT are graft shortage and a recipient’s sensitization to anti-human leukocyte antigens (HLA). In France, about 30% of patients on waiting lists for a KT are sensitized [2]. The number of newly listed patients has increased by 35% over the past 10 years and the number of patients on waiting lists has increased by 82% within 10 years. Pre-existing donor-specific alloantibodies (DSA), defining HLA-incompatible (HLAi) KT, may restrict access to a living-donor transplant or delay access to a deceased-donor KT. Highly sensitized patients remain on a waiting list for two to three times longer than non-sensitized candidates [3].

Options to enable access to KT for sensitized patients include acceptable mismatch programs, paired donation, or desensitization [4]. HLA desensitization significantly improves access to a deceased- or living-donor KT [5]. In 2016, Orandi et al. reported a survival benefit in the USA for sensitized patients undergoing desensitization for HLAi living-donor KT compared to those remaining on a waiting list [6]. 

The goal of desensitization is to reduce DSA mean fluorescence intensity (MFI) as much as possible to obtain a negative cytotoxic crossmatch at the time of KT. Various desensitization protocols have been used in the setting of HLAi KT: most involve plasmapheresis, but also intravenous immunoglobulins and B-cell depleting agents [7]. Plasmapheresis includes several types of extracorporeal therapies that can be used to remove antibodies (anti-HLA antibodies and DSA): plasma exchange (PE), double-filtration plasmapheresis (DFPP), and semi specific immunoadsorption (IA). To date, there is no evidence for superiority of one technique over another and no study has compared the different apheresis techniques in connection to HLA desensitization. The aim of this study was to assess the efficacy, safety and tolerance of each apheresis technique in the setting of desensitization for HLAi KT. 

## 2. Materials and Methods

### 2.1. Study Population

In this single-center study, all adult patients that had undergone desensitization for HLAi KT in the University Hospital of Grenoble, since January 2016, were included. Inclusion into the desensitization protocol required being on the KT waiting list for >3 years, having no infectious or neoplastic co-morbidities, and having optimal results from a cardiac check-up within the previous three months. For living-donor KT, patients were included in case of pre-existing DSA of >1500 MFI. MFI assessment was performed using a bead assay (Luminex Single Antigen assay, Immucor, Norcross, GA, USA). For deceased donors, recipients had to be highly sensitized (i.e., to have a historical calculated panel-reactive alloantibody (cPRA) of ≥80%). The cPRA is calculated as the percentage of HLA antigens out of a panel reacting with the serum of a patient. It represents the percentage of donors expected to react with the serum of the patient. The screening for pretransplant HLA sensitization was also performed by Luminex assay. There were 22 living-kidney and 28 deceased-donor kidney-transplant candidates in this study.

All patients signed an informed consent form. All medical data were collected from our database (CNIL (French National committee for data protection) approval number 1987785v0).

### 2.2. Endpoints

The primary outcome was the efficacy of performing HLAi KT after desensitization and to compare the efficacy to remove HLA antibodies and DSAs between the three apheresis techniques. DSAs were monitored at least once a week during the desensitization period until KT. 

“Intra-session DSA reduction” was defined as the percentage reduction in the immunodominant DSA MFI between pre- and post-apheresis session.

“Inter-session DSA reduction” was defined as the percentage reduction within two consecutive immunodominant DSA MFIs measured before an apheresis session and performed using the same apheresis technique (IA, DFPP, or PE). The number of sessions between two consecutives MFI measures varied but was taken into account within the analyses.

The secondary endpoints were the safety of the apheresis techniques, based on the number of severe adverse events, hemodynamic tolerance, and the evolution of biological parameters (platelet, hemoglobin, leukocytes, fibrinogen). Severe adverse events were defined in this study as occurring during an apheresis session and that led to discontinuing a session or that needed hospitalization. Hypotension was defined as a nadir systolic blood pressure of ≤90 mmHg during apheresis. Technical issues were defined as the need for a nurse’s intervention. 

### 2.3. Procedures

Desensitization and immunosuppression protocols are summarized in Figure 1A for living donors and Figure 1B for deceased donors and was realize in one center. Patients received two rituximab injections (375 mg/m^2^ each). The immunosuppressive regimen consisted of prednisone (0.5 mg/kg), mycophenolate mofetil (500 mg × 2 per day), and tacrolimus (initial dose 0.1 mg/kg/day, with a target trough concentration of between 8 and 10 ng/mL). 

Apheresis sessions were performed by IA, PE, or DFPP according to the initial MFI (s) of DSA (s) for living-donor kidney-recipient KT or according to the immunodominant anti-HLA alloantibody for a deceased donor’s KT. PE or DFPP was performed if MFI was <6000 and IA was performed if MFI was >6000. 

PE was performed by centrifugation using a Spectra Optia^®^ (BCT Lakewood, Terumo, CO, USA) or Comtec^®^ (Fresenius Kabi, France). Filtration was carried out with a Plasmaflo^TM^ OP-08W (Asahi Kasei Medical, Tokyo, Japan). DFPP was equipped with two filters in series. A primary filter with large pores (Plasmaflo^TM^ OP-08W) separated cells and plasma, followed by a specific secondary filter (Cascadeflo^TM^ EC-20W) for immunoglobulin filtration. IA was performed after plasma centrifugation on two adsorber Globaffin^®^ columns (Fresenius Medical Care, St. Wendel, Germany) working in tandem. IA could be coupled with membrane filtration (Monet^®^). Monet^®^ filter was used to enhance the removal of molecules possibly involved in the post-transplantation rejection risk such as IgM, C1q, properdin, mannose-binding lectin [8]. It was associated with IA when HLA antibody titer was high (i.e., >12,000). All patients received prophylactic antibiotherapy with phenoxy-methylpenicillin and sulfamethoxazole-trimethoprim. Apheresis sessions were carried out in parallel with the hemodialysis sessions. Intravenous Immunoglobulins (IVIg) were given at low dose (1.5 g) in case of low IgG level (<4 g/L) before the apheresis session for substitution purposes. 

### 2.4. Desensitization Protocol

For living-donor KT, the protocol consisted of four or five apheresis sessions per week for 2 weeks prior to KT. If the DSA MFI was >12,000, IA was performed daily. If DSA MFI was <6000, IA could be replaced by DFPP or PE to achieve a threshold MFI of < 3000 before KT. KT was performed when DSAs had an MFI of <3000, i.e., a negative-flow cytometric crossmatch in our center on the day before KT. A systematic graft biopsy was performed at 1, 3 and 12 months.

For deceased-donor KT, three to five apheresis sessions per week were carried out until a compatible kidney graft was available. If no nationally available graft was proposed within 45 days after starting desensitization, the first local ABO-compatible graft, matched for age and weight, was proposed. To facilitate the purification of high MFIs HLA antibodies, some patients with high level of antibodies (MFI > 12000) and waiting for a deceased donor were primed by receiving tocilizumab injections before the start of apheresis [9]. 

### 2.5. Statistical Analyses

Quantitative data are presented as means ± standard deviations (SD), or as medians with quartiles (Q1–Q3). Qualitative data are presented as the numbers of patients and percentages. The chi-squared test was used for categorical variables; the Wilcoxon or the Kruskal—Wallis test was used for continuous variables. Multiple linear-regression analysis was performed to identify the independent factors associated with inter-session and intra-session immunodominant DSA evolution. All parameters significantly associated with immunodominant DSA inter-session and intra-session decrease were included in the multivariate analyses except for the number of sessions that was closely correlated to the total volume of treated plasma and did not provide additional relevant data. Data adjusted in the multivariate inter-session model were the total volume of treated plasma and the type of apheresis technique (ie PE, DFPP and IA). Data adjusted in the multivariate intra-session model were the total volume of treated plasma, the type of apheresis technique and the initial MFI of the immunodominant DSA. In order to assess the impact of patient’s variability on DSA reduction, we used a mixed model that allowed to predict the fixed effect and variability of the apheresis. A two-sided *p*-value of < 0.05 was considered statistically significant. Statistical analyses were conducted using R statistical software. 

## 3. Results

### 3.1. Study Population

Between August 2016 and November 2020, 45 patients were desensitized in the setting of HLAi KT at Grenoble University Hospital (Table 1). Patients were aged 53 ± 13 years, and 25 (55.6%) were women. Mean body-mass index was 24 ± 4 kg/m^2^. Seventeen patients (44%) were desensitized in the setting of a living-donor HLAi KT. Among these, eight were also ABO incompatible. Mean cPRA was 84.6 ± 26.3% (96 ± 5% for deceased donors and 65 ± 35% for living donors). A total of twenty-three (59%) had undergone a previous KT and median time on dialysis before desensitization was 65 (16.5–110) months. 

At the beginning of desensitization, for living donors, the median immunodominant class I DSA MFI was 6195 (2458–11,347) and was 2191 (1180–7238) for class II DSAs. The deceased-donor median for immunodominant class I DSA MFIs was 13,929 (5237–18,606) and was 5508 (2079–10,872) for class II DSA. Retrospectively, 27 (60%) patients had more than one DSA. Regarding class I DSAs, anti HLA-A was present in 77.7% of patients, anti HLA-B in 63% of patients, and anti HLA-C in 15%. Regarding class II DSAs, anti HLA-DP was present in 28% of patients, anti-HLA-DQ in 36%, and anti HLA-DR in 44%. 

### 3.2. Characteristics of Apheresis for HLAi Kidney Transplantation

Between January 2016 and January 2020, 881 apheresis sessions were carried out for the 45 patients in the setting of HLA-incompatible KT. The characteristics of all apheresis sessions are summarized in Table 2. The number of sessions per patient was 15 [10,11,12,13,14,15,16,17,18,19,20,21,22,23,24]. IA was the most performed technique with 720 (81.7%) sessions. The median duration between the first and last session for each patient was 29 (15–51) days. The median duration of one session was 3.2 h (2.6–3.9): IA sessions took significantly longer (3.5 ± 0.8 h) compared to DFPP (2.1 ± 0.6 h) and PE (1.9 ± 0.6 h) (*p* < 0.001). Each patient had 9 ± 6 IA sessions, 2 ± 1 PE sessions, and 3 ± 2 DFPP sessions. The Monet^®^ filter was added in 340 IA sessions (47.2%). A total of thirteen patients (28.9%) had received at least one injection of tocilizumab prior to apheresis desensitization at a dose of 8 mg/kg. A total of nineteen (42.2%) patients received IVIg injections in 47 IA sessions (6.5%) at the dose of 140 mg/kg, i.e., a mean dose of 9.5 ± 7 g Fibrinogen was infused after 51 sessions at a mean dose of 1.8 ± 0.8 g. 

### 3.3. Efficacy of Apheresis and Access to Kidney Transplantation

Regarding assess to KT, 39 (87%) patients received an HLAi KT at post-desensitization. A total of six desensitized patients did not receive a transplant: this was because of failure to remove HLA antibodies from three patients (6.6%) or intercurrent events occurring during the desensitization period for the other three patients (6.6%). The intercurrent events were one myocardial infarction (with death), a pulmonary infection (pneumocystis), and a digestive perforation. One patient died during the desensitization protocol period from acute coronary syndrome. The number of sessions was associated with the MFI level of the immunodominant DSA before the desensitization. An MFI increase of 276 of the DSA before the desensitization procedure was associated with an additional session needed to access to KT (*p* < 0.001).

We then assessed factors associated with intra- and inter-session DSA evolution. 

Intra-session analyses: For class I DSAs, the mean decrease of MFI was 13 ± 4%. In univariate and multivariate analyses, the volume of purified plasma was significantly associated with a higher decrease in intra-session MFI (*p* = 0.03). For class II DSAs, the mean decrease in MFIs was 83 ± 22%. In univariate and multivariate analyses, IA and PE, and a lower initial DSA MFI were significantly associated with a higher decrease in intra-session MFI (*p* < 0.01) (Table 3). The mixed model used to predict patient variability impact on the apheresis effect showed similar results to the previous model meaning that the patient variability did not significantly impacted the antibodies removal (Appendix A).

Inter-session analyses: For class I DSAs, the mean decrease in MFI was 88 ± 50%. In univariate and multivariate analyses, the volume of treated plasma and the IA were associated with a higher inter-session DSA decrease, *p* = 0.04 and *p* = 0.03, respectively. For class II DSAs, the mean decrease in MFI was 59 ± 34%. In univariate and multivariate analyses, a higher total volume of treated plasma was significantly associated with a decrease in inter-session MFI (*p* = 0.02) (Table 4). 

The efficacies of the intra-session subtype immunoglobulin reduction are summarized in Table 4. The best reduction rate of IgG was −60.6% (−46; −73) for PE sessions, followed by −60% (−33; −69) for IA, and −40.0% (−30; −50) for DFPP (*p* < 0.001). Figure 2 shows the percentages of IgG reduction according to apheresis techniques. The volume of treated plasma is significantly associated with IgG reduction post apheresis for all techniques, but IA needs a more important volume to remove Ig. The absolute value of IgG at post-session was lower for IA (0.6 ± 0.7 g/L) versus EP (2.1 ± 1 g/L) and DFPP (1.4 ± 1 g/L), *p* < 0.001. The use of the Monet filter was associated with a significantly higher reduction of IgG but also IgM and IgA post session as compared to IA alone (Appendix A). 

### 3.4. Apheresis Tolerance

Clinical tolerance: serious adverse events occurred in 17 (1.9%) sessions and hemodynamic intolerance occurred in 154 (17.5%) sessions. We assessed the association of serious adverse events with age, trough level of tacrolimus, technique of apheresis, Rituximab, IVIg, Tocilizumab, simultaneous dialysis, vascular access, use of membranous filter, duration of apheresis session, anticoagulation and blood flow rate. DFPP was significantly less well-tolerated compared to IA and PE: serious adverse events occurred in 6.5% of DFPP sessions versus 1.9% and 1.2% for PE and IA, respectively (*p* < 0.01). Trough level of tacrolimus was also associated with serious adverse events (*p* = 0.02). Intrasession hypotension occurred in 39.3% of DFPP sessions versus 20.4% and 14% for PE and IA, respectively (*p* < 0.01). The number of sessions with technical issues that required a nurse’s intervention was 88 (10%) and was similar between the three techniques (*p* = 0.53). 

Biological tolerance (Table 5): fibrinogen decreased by −46.7% (−23; −60) with a higher loss with DFPP: -1.5% (−55; −69) versus PE −33.3% (−28; −64) and IA −42.9% (−22; −57) (*p* < 0.01). Post-session fibrinogen was lower with DFPP: 0.6 ± 0.4 g/L compared to the other techniques (1.0 ± 0.7 g/L for IA and 1.3 ± 0.7 g/L for PE) (*p* < 0.01). Five (11.1%) patients presented with asymptomatic cytomegalovirus (CMV) DNAemia and 9 (20%) with Epstein—Barr virus (EBV) DNAemia during the desensitization period. Only one patient developed CMV disease with digestive involvement. Red-blood cell transfusion was performed in 82 (9.3%) sessions. 

## 4. Discussion

In this cohort, we found that an MFI-stratified apheresis protocol associated with rituximab and a standard immunosuppressive regimen was efficient to desensitize patients in the setting of HLAi KT. Only six patients did not receive a transplant due to failure of desensitization or an intercurrent event. Removal of intra-session class II DSAs was more efficient with IA and PE than with DFPP and when the maximal DSA was lower. The decrease in inter-session class I and II DSAs was associated with the higher volume of treated plasma (in multivariate analyses). IA was also associated with a better class I inter-session decrease.

The very first plasmapheresis technique was performed on dogs in 1914 [10]. In CKD and KT, the main pathologies associated with plasmapheresis are antibody-mediated rejection, focal segmental glomerulosclerosis, and desensitization [11]. Highly sensitized patients without a HLA-compatible donor are difficult to manage. These patients have to wait long periods for a compatible deceased donor, are often on hemodialysis, and have increased morbidity-mortality [12]. Desensitization strategies have significantly improved access to KT from deceased and living donors [5,13]. 

The goal of desensitization is to obtain a sustained drop in DSA MFI and to allow KT under an acceptable immunological risk. In the 1970s, Cardella et al. considered that PE could remove DSAs involved in acute humoral rejection [14]. In desensitization, plasmapheresis has shown better results compared to IVIg to achieve a negative crossmatch and was associated with a lower rate of antibody-mediated rejection [15]. Anti-HLA antibodies are IgG, with a half-life of 21 days, with a molecular weight about 160,000 Daltons, and a vascular distribution of about 40% [16]. For these reasons, apheresis can remove DSAs from plasma. 

To date, there are three different apheresis techniques: PE, DFPP, and IA. PE is a non-selective technique that removes all plasma proteins. In our center, PE has been mostly performed by centrifugation, which allows a decrease in blood flow and reduces the session time [17]. DFPP and IA are more recent techniques that allow selective or semi-selective plasma purification. Selective plasma purification avoids the unnecessary loss of plasma proteins and reduces the need for liquid replacement and increases the efficiency of purification [18].

Böhmig et al., in 2007, showed, in a randomized study, the efficacy of IA to remove antibodies in a setting of acute antibody-mediated rejection post KT [19]. In our study, the most effective apheresis technique was IA. The first use of IA in KT for highly HLA-sensitized cases dates from 1989 [20]. IA has since been used successively as a desensitizing therapy by many teams [21]. The most commonly used IA column in our center has been a Globaffin^®^ column: it uses a synthetic peptide with a high affinity for the constant fraction (Fc) of IgG antibodies of subclasses 1, 2 and 4 [22]. By purifying a high plasmatic volume with IA, Belàk et al. have shown a 87% drop in the initial IgG level and good affinity for IgG 2 and 4 [23]. IA and DFPP allow higher plasma volumes to be treated without excessive loss of plasma [24,25] whereas the main constraint of PE remains the necessary use of a substitute solution. 

To the best of our knowledge, only a few studies with very small populations have compared apheresis techniques in the setting of HLAi KT [23,26]. In our study, the percentage IgG reduction was higher with PE, but the absolute value of IgG at post-session was lower with IA. This is partly due to the pore size of DFPP filters, which allows good elimination of IgA and IgM, but low elimination of IgG to avoid loss of albumin with a similar molecular weight [27]. Regarding to HLA antibodies removal and monitoring, the limitation is the measurement itself by Single-Antigen Bead assay. Indeed, high level of HLA antibodies can be missed or underestimated because of IgG detection interference (prozone effect). In order to prevent this, all patients of this study had a dilution test of their serum before the desensitization procedure and none had a prozone effect. Moreover, in our study, the initial MFI of HLA antibody was not similar for IA, PE and DFPP. We may suspect that the reduction of MFI is partially correlated with the amount of antibody which introduce of possible bias in our results.

Apheresis requires both medical and paramedical expertise with a team that is well-trained in the different techniques. Plasmapheresis may be complicated by cardiovascular [28], hemorrhagic [29,30] or allergic [31] complications. In our center, the technique with the most undesirable effects was DFPP. We also found a significant increase in the numbers of leukocytes after DFPP. This may be explained by the bio-incompatibility of the membranes and the frictional forces imposed on blood through these membranes [32]. This activation of the inflammatory system may be partly responsible for the excess risk of hypotension. 

Moreover, even if not assessed in this study, we suspect there is improved quality of life for patients that receive a transplant after desensitization compared to those that remain on dialysis.

Finally, desensitization with apheresis was effective at removing HLA antibodies and allowed access to HLAi KT for sensitized patients. IA and EP were more effective to remove IgG and antiHLA antibodies, especially for class II DSAs, and were better tolerated than DFPP.

## Figures and Tables

**Figure 1 jcm-10-01316-f001:**
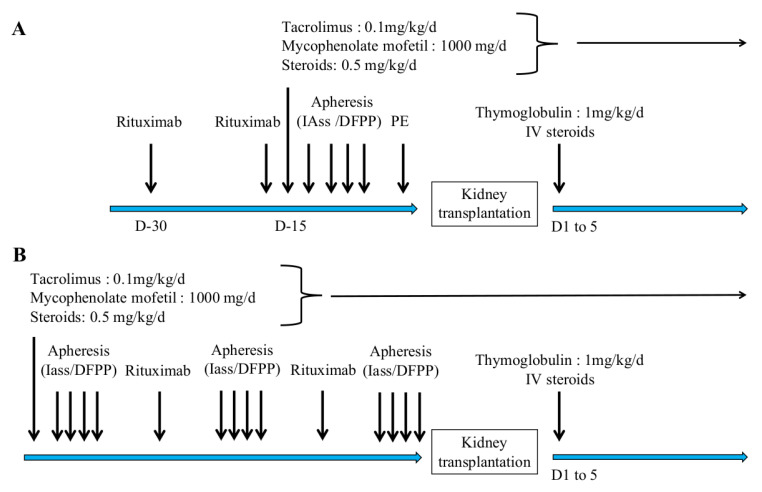
Desensitization and immunosuppression protocol for HLA-incompatible kidney transplantation. Panel (**A**) shows the protocol for living donors HLAi kidney transplantation. Panel (**B**) shows the protocol for deceased donors HLAi kidney transplantation. IAss: semi-specific immunoadsorption; DFPP: double-filtration plasmapheresis; PE: plasma exchange.

**Figure 2 jcm-10-01316-f002:**
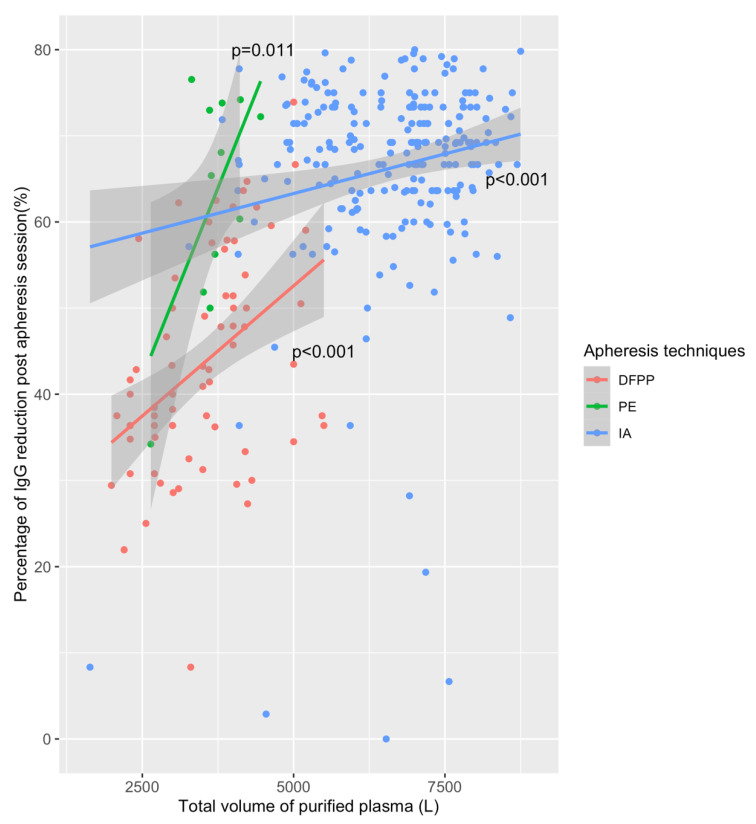
Post session reduction of immunoglobulin-G according to the apheresis technique. DFPP: double-filtration plasmapheresis; PE: plasma exchange; IA: immunoadsorption; IgG: immunoglobulin subtype G. IgG reduction was assessed in all sessions that did not received IV immunoglobulins. The volume of purified plasma is significantly associated with IgG reduction.

**Table 1 jcm-10-01316-t001:** Baseline characteristics of patients according to kidney transplant donor type.

	Desensitization with Living Donors	Desensitization with a Total of Deceased Donors	Total	*p*-Value
	*N* = 18	*N* = 27	*N* = 45	
Age at inclusion (years)	53.6 ± 15	51.9 ± 12	52.6 ± 13	0.84
Male/Female gender ratio	7/11	13/14	20/25	0.54
Body Mass index (Kg/m^2^)	24.5 ± 3	24.2 ± 4	24.3 ± 4	0.54
History of previous transplantation—*N* (%)	6 (35.3)	17 (77.3)	23 (59)	0.02
Pre-emptive kidney trans—*N* (%)	1 (5.9)	0 (0)	1 (2.6)	0.25
Time on dialysis (months)	17 ± 15	150 ± 114	92 ± 108	<0.001
cPRA (%)	65 ± 35	96 ± 5	84 ± 26	<0.001
>1 class I DSA—*N* (%)	11 (65)	16 (59)	27 (60)	0.09
>1 class II DSA—*N* (%)	12 (66.6)	13 (48)	25 (55.5)	0.01
Number of class I	17 KT	22 KT	39 KT	0.31
Missmatch—*N* (%)				
1	1 (5.8)	0 (0)	1 (2.5)	
2	7 (41)	6 (27)	13 (33.3)	
3	8 (47)	9 (41)	17 (43.5)	
4	1 (5.8)	5 (22.7)	6 (15.3)	
Number of class II				0.93
Missmatch—*N* (%)				
0	3 (17.6)	3 (13.6)	6 (15.3)	
1	3 (17.6)	5 (22.7)	8 (20.5)	
2	7 (41)	8 (36.3)	15 (38.4)	
3	2 (11.7)	3 (13.6)	5 (12.8)	
4	2 (11.7)	1 (4.5)	3 (7.7)	
Mean number of PE sessions	2 ± 1	1 ± 1	1 ± 1	0.04
Mean number of DFPP sessions	3 ± 3	2 ± 3	2 ± 3	0.61
Mean number of IA sessions	6 ± 7	22 ± 17	16 ± 15	<0.001
Trough tacrolimus level (ng/mL) at inclusion	4.9 ± 0.5	9.8 ± 5.8	8.5 ± 5.5	0.02

cPRA: calculated Panel Reative Antigen, DSA: Donor specific antibody, DFPP: double-filtration plasmapheresis; PE: plasmatic exchange; IA: immunoadsorption.

**Table 2 jcm-10-01316-t002:** Characteristics of sessions according to apheresis technique.

	DFPP (*N* = 107)	PE (*N* = 54)	IA (*N* = 720)	Total (*N* = 881)	*p*-Value
Duration of session (hours). Median [IQ]	2 (1.8–2.3)	1.7 (1.5–2.0)	3.5 (2.9–4.0)	3.2 (2.6–3.9)	<0.01
Treated plasma volume (L). Median [IQ]	3675 (3000–4200)	4200 (2564–3685)	6641 (5520–7523)	6035 (4803–7286)	<0.01
Blood flow (mL/min) Mean ± SD	146 ± 14	63 ± 38	54 ± 7	65 ± 32	<0.01
Substitution volume (L) Mean ± SD	259 ± 224	2857 ± 750	104 ± 53	295 ± 717	<0.01
Substitution fluid *N* (%)					<0.01
– Albumin 20%	10 (9.3%)	0 (0%)	720 (100%)	730 (83%)	
– Albumin 20% + saline serum	88 (82.2%)	0 (0%)	0 (0%)	88 (10%)	
– Albumin 4%	9 (8.4%)	20 (37%)	0 (0%)	29 (3.2%)	
– FFP	0 (0%)	34 (63.0%)	0 (0%)	34 (3.9%)	

DFPP: double-filtration plasmapheresis; PE: plasmatic exchange; IA: immunoadsorption; FFP: fresh frozen plasma.

**Table 3 jcm-10-01316-t003:** Uni- and multivariate analyses of factors associated with reduction in intra-session MFI of immunodominant DSAs.

	**DSA Class I**		**DSA Class II**	
	**Univariate *p*-Value**	**Multivariate *p*-Value**	**Univariate *p*-Value**	**Multivariate *p*-Value**
Volume of treated plasma	0.01	0.03	0.07	0.37
Technique of apheresis	0.20	0.60 (IA)	0.02 (PE)	<0.01 (PE)
(IA and PE vs. DFPP)			0.01 (IA)	<0.01 (IA)
Duration of session	0.89	-	0.10	-
Maximum MFI of DSA	0.39	0.19	<0.01	<0.01

DFPP: double-filtration plasmapheresis; PE: plasmatic exchange; IA: immunoadsorption; DSA: donor-specific antibody; MFI: mean fluorescence intensity.

**Table 4 jcm-10-01316-t004:** Uni- and multivariate analyses of factors associated with reduction of inter-session MFI of immunodominant DSAs.

	DSA Class I		DSA Class II	
	Univariate *p*-Value	Multivariate *p*-Value	Univariate *p*-Value	Multivariate *p*-Value
Volume of treated Plasma	0.24	0.04	0.06	0.02
Technique of apheresis	0.86 (PE)	0.83 (PE)	0.22 (PE)	0.18 (PE)
(IA and PE vs. DFPP)	0.11 (IA)	0.03 (IA)	0.76 (IA)	0.38 (IA)
Delay between sessions	0.42	-	0.49	
Duration of session	0.92	-	0.78	-

DFPP: double-filtration plasmapheresis; PE: plasmatic exchange; IA: immunoadsorption.

**Table 5 jcm-10-01316-t005:** Biological parameters according to apheresis techniques.

	DFPP (*N* = 107)	PE (*N* = 54)	IA (*N* = 720)	Total (*N* = 881)	*p*-Value
Pre-post IgA evolution (%) Median [IQ]	−55 (−45; −63)	−48 (−1; −71)	−14 (−7; −21)	−17 (−8; −29)	<0.01
Pre-post IgG evolution (%) Median [IQ]	−40 (−31; −50)	−61 (−46; −73)	−60 (−33; −70)	−56 (−33; −69)	<0.01
Pre-post IgM evolution (%) Median [IQ]	−37 (0; −58)	−51 (60; −75)	−17 (0; −54)	−17 (0; −57)	0.10
Pre-post Alb evolution (%) Median [IQ]	1 (14; 2)	10 (14; −3)	9 (14; −1)	9 (15; 0)	0.73
Pre-post fibrinogen evolution (%) Median [IQ]	−61 (−56; −69)	−33 (−29; −64)	−43 (−22; −57)	−47 (23; −60)	<0.01
Pre-post hemoglobin evolution (%) Median [IQ]	15 (22; −8)	2 (10; −2)	2 (9; −2)	3 (11; −2)	<0.01
Pre-post leukocytes Evolution (%) Median [IQ]	65 (96; 33)	22 (60; 5)	4 (18; 8)	8 (27; 6)	<0.01
Pre-post platelet evolution (%) Median [IQ]	7 (−1; 17)	14 (2; 21)	12 (3; 21)	12 (2; 21)	0.01

DFPP: double-filtration plasmapheresis; PE: plasmatic exchange; IA: immunoadsorption.

## Data Availability

The data presented in this study are available on request from the corresponding author.

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
