# Peer review of "Apheresis Efficacy and Tolerance in the Setting of HLA-Incompatible Kidney Transplantation"

_jcm, 2021, doi:10.3390/jcm10061316_

Round 1
Reviewer 1 Report
Thank you for the opportunity to review this paper. This reviewer believes evaluate Apheresis efficacy and tolerance in the setting of HLA-incom-2 patible kidney transplantation is important and publication-worthy.
There are few concerns of this manuscript about the statistical method.
- Will patients’ demographics and other comorbidities impact the performance of apheresis technique?
- As stated by researchers, “881 apheresis sessions were carried out for 164 of the 45 patients.” So when researchers analyzed their data, did they adjust patients' ID? Would there be a patient receiving more than one type of apheresis technique? If yes, would that make a difference if researchers adjust patients’ ID and their baseline characteristics?
- What covariates had been adjusted in the multivariate model in table 2?
Author Response
We agreed with this relevant remark.
Indeed, 881 apheresis were carried out for 45 patient (not 164) and all sessions were included in the analyses.
We have redone the analyses, taking this remark into account:
Regarding to the covariates: we decided to remove the parameter “number of sessions” for the analyses of intersession DSA reduction because it was very correlated to the volume of treated plasma (see Figure below).
Regarding to the adjustment of patients’ ID:
We adjusted a random effect for patients, modeling the effect of session type as a fixed and a random effect (nested within patients), in a mixed model. This allows to predict the fixed (mean) effect of the apheresis type, and the variability around that fixed effect (due to inter-patient’s variability).
In this mixed model, the parameters estimation remained very close and there was no change in significance (tables below): there was no effect of session type when accounting for inter-patient variability.
We modified the statistical paragraph and results paragraph according to the new multivariate model. We did not add the mixed model into the manuscript because the think that it makes the message more complex but we added it in a supplementary table at the end of the manuscript.
|
Fixed model
|
Mixed model
|
Variance ± SD |
||
Inter-session MFI of immunodominant class I DSAs |
Plasma exchange Immunoadsorption Volume of treated plasma |
p=0.829 p=0.035 p=0.047 |
Plasma exchange Immunoadsorption Volume of treated plasma |
p=0.829 p=0.035 p=0.047 |
1x 10-3 ± 0.03 3x 10-4 ± 0.01
|
|
Fixed model
|
Mixed model
|
Variance ± SD |
||
Inter-session MFI of immunodominant class II DSAs |
Plasma exchange Immunoadsorption Volume of treated plasma |
p=0.180 p=0.388 p=0.022 |
Plasma exchange Immunoadsorption Volume of treated plasma |
p=0.195 p=0.375 p=0.018 |
133 ± 11 22 ± 5 |

Reviewer 2 Report
In this interesting study, Noble and coworkers report on their impressive single-center experience with the use of different apheresis techniques in the context of transplant recipient desensitization. This is a very well written and detailed report of immunological/clinical outcomes obtained in a considerable number of patients (45 subjects; 881 apheresis sessions). Of substantial interest is the comparative analysis of different techniques. The study may be of high interest for the transplant community.
I have only a few minor points:
The authors should explain definitions of cPRA in more detail (all Ab included, or preselection according to CDC testing and/or Luminex thresholds?).
The authors should explain the indication for including Monet filters, and discuss the respective literature. Did this filter affect MFI outcomes? If data are available, the effect on IgM vs IgG levels would be of interest.
Using DSA-MFI as endpoint, the authors should include a critical discussion of prospects and limitations of this parameter, which may not accurately reflect the level of a given antibody. They should also discuss the use of dilution experiments as a potential strategy to facilitate the interpretation of data.
Author Response
1)
-For deceased donors, the inclusion criteria were only based on the cPRA level above 80 %. We assumed that only highly sensitized patients without a living donor kidney transplantation perspective were ideal candidates.
-For living donors, the inclusion criteria were the presence of at least one DSA assessed by a single antigen bead assay (Luminex Single Antigen assay, Immucor, USA). In our center, the threshold of DSA positivity is 1500.
We included the definition of cPRA and we reworded the study population section in the manuscript page 3-4 to be more accurate on this point.
2)
We have conducted several works on the adjunction of a membrane filtration to Immunoadsorption. Monet filter is added because of its potentiating effect to remove IgM, C1q, properdin and MBL proteins (Defendi et al. transplant international, 2019, PMID: 30901502). The reviewer is right to suggest that the adjunction of this membrane filtration may impact the outcome. We also shown that Monet filter is associated with a stronger modification of hemostasis, especially fibrinogen and Factor XIII depletion (Marlu et al. Journal clinical apheresis, 2020, PMID: 32805070).
We added as suggested a comment in the manuscript, page 4, and a reference to explain the use of Monet filter.
The intra session reduction of IgM, IgA and IgG was significantly higher when the Monet filter was used as compared to IA alone (see Figure below). We added the data in a supplementary figure so as not to interfere with the main message of this article.
3)
We totally agree with this comment.
MFI measurement using Single Antigen bead assays is imperfect. High level of HLA antibodies can be missed or underestimated in single antigen bead assay because of IgG detection interference. In order to prevent this prozone effect, all patients had a dilution test of their serum before the desensitization procedure.
This information was missing in our manuscript, so we added it in the discussion section as suggested.

Reviewer 3 Report
Overall this is a very valuable study the authors should be complemented for. Noble et al describe a unique approach of an apheresis-centred, situation-adaptive desensitization protocol that allowed for kidney transplantation in up to 90% of highly sensitised patients on the waiting list.
There are some minor remarks that once be addressed by the authors do not hinder publication of this overall very good work.
- There are some phrases that require careful proof-reading to eliminate grammatical errors (e.g. 3rd person's s in the first sentence of the abstract).
- Please provide Baseline Characteristics of the patients within a table comprising information (als already noted in the text!) on HLA DSA against class I and II, and seperately on A B C DQ DR (and DP if availabe), both in number of patients and in MFI, the binding strength of the immunodominant DSA, number of sessions per patient and for each type of sessions, additional immunuosuppression - of course besides standard variables such as age, sex, time on dialysis, cause of ESRD, Number of prior Tx, vPRA, PRA, percentage of plausible DSA with known immunization event, UAGs, mean Tac trough levels, HLA-MM, donor age, etc.
This should be done for both of the protocols seperately and also over.- Please mention that not only DSA directed against HLA but also against non-HLA may be removed by either of the techniques.
- Please discuss the MFI is not a linear representation of DSA affinity and concentration. Titration experiments show, that a drop in high MFI levels b
- Please provide additional information in lines 82-82 how adjustment for inter-session DSA reduction according to number and type of intercurrent treatment sessions was done. This is not clear. If adjustment is not possible consider omitting the analysis.
- Please provide more information on how DSA binding strength affected time to Tx in the DD cohort.
- Please provide information on the instances when additional immunosuppressive therapy was used (such as tocilizumab / IvIG)
- Please provide additional information in which cases Membrane filtration was added.
- When discussing the evolution of IAS in KTx please consider citing the only randomised study (Böhmig et al Jasn 2007)
- Table 2 - if I understood correctly it is always an univariate (=reduction of MFI) but either an univariable or multivaribale anlyses. Please double check. And correct also eg. lines 189, 202, 244 and others.
The same goes for Table 3.
- Please provide a figure legend fpr Figure 2. Please explain in the text more carefully what Figure 2 shows. It is very counterintuitive the the treated plasma volume does not impact on IgG reduction only in IAS. This should be discussed within in the text and analysed with caution.
- Please be more extensive concerning adverse events, do not limit yourself to intrasession adverse events and consider providing an additional table stratified by number of sessions and additional immunosuppression (tocilziumab) including infectious complications etc. Where there any cases of diverticulitis?
- Please carefully discuss the impact of the initial strength of DSA on the reduction as measured by MFI levels. This may be a strong confounder when comparing PE and IAS given the dysbalance of DSA MFI within the two techniques.
Author Response
-We corrected some grammatical errors
-We added a table 1 of patients baseline characteristics.
|
Desensitization with Living donors N=18 |
Desensitization with Deceased donors N=27 |
Total N=45 |
p-value |
Age at inclusion (years) |
53.6 ± 15 |
51.9 ± 12 |
52.6 ± 13 |
0.84 |
Male/Female gender ratio |
7/11 |
13/14 |
20/25 |
0.54 |
Body Mass index (Kg/m2) |
24.5 ± 3 |
24.2 ± 4 |
24.3 ± 4 |
0.54 |
History of previous transplantation – N(%) |
6 (35.3) |
17 (77.3) |
23 (59) |
0.02 |
Pre-emptive kidney trans – N(%) |
1 (5,9) |
0(0) |
1(2,6) |
0.25 |
Time on dialysis (months) |
17 ± 15 |
150 ± 114 |
92 ± 108 |
<0.001 |
cPRA (%) |
65 ± 35 |
96 ± 5 |
84 ± 26 |
<0.001 |
> 1 class I DSA – N(%) |
11 (65) |
16 (59) |
27 (60) |
0.09 |
> 1 class II DSA – N(%) |
12 (66.6) |
13 (48) |
25 (55.5) |
0.01 |
Number of class I Missmatch -N(%) 1 2 3 |
17 KT
1 (5.8) 7 (41) 8 (47) 1 (5.8) |
22 KT
0 (0) 6 (27) 9 (41) 5 (22.7) |
39 KT
1 (2.5) 13 (33.3) 17 (43.5) 6 (15.3) |
0.31 |
Number of class II Missmatch -N(%) 0 1 2 3 |
3 (17.6) 3 (17.6) 7 (41) 2 (11.7) 2 (11.7) |
3 (13.6) 5 (22.7) 8 (36.3) 3 (13.6) 1 (4.5) |
6 (15.3) 8 (20.5) 15 (38.4) 5 (12.8) 3 (7.7) |
0.93 |
Mean number of PE sessions |
2 ± 1 |
1 ± 1 |
1 ± 1 |
0.04 |
Mean number of DFPP sessions |
3 ± 3 |
2 ± 3 |
2 ± 3 |
0.61 |
Mean number of IA sessions |
6 ±7 |
22 ± 17 |
16 ± 15 |
<0.001 |
Trough tacrolimus level (ng/mL) at inclusion |
4.9 ± 0.5 |
9.8 ± 5.8 |
8.5 ± 5.5 |
0.02 |
-We added a comment on the efficacy of the plasmapheresis techniques to remove antiHLA antibodies and also donor specific antibodies in the introduction section page 3.
-We added a comment on the limitation of the MFI measurement and the risk of Prozone effect mentioned by the reviewer. This is a limitation in the DSA and anti-HLA monitoring. All patients had a dilution test before the inclusion in the desensitization protocol.
- Data were adjusted in the multivariate intra-session model with the total volume of treated plasma, the type of apheresis technique and the initial MFI of the immunodominant DSA. In order to assess the impact of patient’s variability on DSA reduction, we used a mixed model that allowed to predict the fixed effect and variability of the apheresis. We reworded the manuscript with this sentence to be more accurate in the statistical analyses section page 6.
-We assessed the correlation of MFI level (of the immunodominant DSA) and the number of apheresis session. An MFI increase of 276 was associated with an additional session (p<0.001). This answer the question of the correlation to the time to KT access. We added this information in the manuscript page 8.
-We added the precise indications of Tocilizumab and IvIG in the material section page 5 and 6. Ivig were not given at an immunomodulation dose but at a substitution dose (1.5g) in case of hypoIgG < 4g before the session.
- Regarding to the membrane filtration, we added some precisions in the procedure section page 5.
- We added the citation of Bohmig in the AJT 2007 in the discussion section page
-We redid the multivariate analyses by changing our model (Cf. statistical analyses and tables). We decided to remove the parameter “number of sessions” for the analyses of intersession DSA reduction because it was very correlated to the volume of treated plasma.
-Regarding to Figure 2. We redid the analyses by removing all sessions that received IVIG because the level if IgG post session was biased. Doing so, we showed that for all apheresis techniques (IA, PE and DFPP), the percentage of IgG reduction was significantly associated with the volume of purified plasma. We modified the manuscript page 9 and changed the figure page 10.
-We performed analyses of severe adverse events according to various factors: age, Trough level of TAC, technique of apheresis, use of Rituximab, IvIG, Tocilizumab, simultaneous dialysis, vascular access, use of membranous filter, duration of session, type of anticoagulation and blood flow rate. The only factors associated with severe adverse events were the DFPP (p=0.001) and the trough level of tacrolimus (p=0.022). We included those results in the manuscript. No cases of diverticulis occurred.
-We insisted in the limitation of DSA comparability in the discussion section page 13.

Round 2
Reviewer 1 Report
The present paper address an interesting and clinical important question, and uses a well thought out method to address it. Review’s previous comments had been well addressed/answered.